# SOBOLEV TRAINING OF END-TO-END OPTIMIZATION PROXIES

## ABSTRACT

Optimization proxies—machine-learning models trained to approximate the solution mapping of parametric optimization problems in a single forward pass—offer dramatic reductions in inference time compared to traditional iterative solvers. This work investigates the integration of solver sensitivities into such end-to-end proxies via a Sobolev–training paradigm and does so in *two distinct settings*: (i) *fully supervised* proxies, where exact solver outputs and sensitivities are available, and (ii) *self-supervised* proxies that rely only on the objective and constraint structure of the underlying optimization problem. By augmenting the standard training loss with directional-derivative information extracted from the solver, the proxy aligns both its predicted solutions *and* local derivatives with those of the optimizer. Under Lipschitz-continuity assumptions on the true solution mapping, matching first-order sensitivities is shown to yield uniform approximation error proportional to the training-set covering radius. Empirically, different impacts are observed in each studied setting. On three large Alternating Current Optimal Power Flow benchmarks, supervised Sobolev training cuts mean-squared error by up to 56 % and the median worst-case constraint violation by up to 400 % while keeping the optimality gap below 0.22 %. For a mean–variance portfolio task trained without labeled solutions, self-supervised Sobolev training halves the average optimality gap in the medium-risk region (i.e. standard deviation above $10\%$ of budget) and matches the baseline elsewhere. Together, these results highlight Sobolev training—whether supervised or self-supervised—as a path to fast, reliable surrogates for safety-critical, large-scale optimization workloads.

## 1 INTRODUCTION

*Optimization proxies* Van Hentenryck (2025); Amos et al. (2023); Donti et al. (2021) are learned functions that emulate the solution operator of a parameterized optimization problem,

$$g(p) = \arg\min_x \left\{ f(x;p) \mid c(x;p) = 0, \ x \geq 0 \right\} = x^*, \hat{g}_\theta(p) \approx g(p), \quad \hat{g}_\theta : p \mapsto \tilde{x} \approx x^*.$$

thereby replacing an iterative solver with a single forward pass, where $p \in \mathbb{R}^d$ denotes the problem parameters, $x \in \mathbb{R}^n$ the decision variables, and $f, c$ are differentiable functions. Such surrogates enable millisecond-scale inference in time-critical domains—real-time grid dispatch, on-device resource allocation, quantitative finance Chen et al. (2022)—and, when embedded inside larger decision pipelines, they provide differentiable "inner loops" that allow an outer optimization to steer itself with respect to problem parameters via the sensitivities $\nabla_p \hat{g}_\theta$. Here, inaccurate sensitivities can lead to slow convergence and poor performing solutions in such nested workflows.

This study explores the idea of jointly learning solution values with a model architecture that approximates the geometric sensitivities of the problem in optimization proxies. It adopts a Sobolev-style loss that supplements value regression with first-order information, an idea first explored for generic function approximation in Czarnecki et al. (2017) and later for approximating functions embedded inside optimization problems Tsay (2021). For optimization problems, the sensitivity analysis of the solutions can be automatically extracted based on methods appropriate to the underlying problem class Pacaud (2025). Expanding on prior literature, the present work makes four main contributions:

1. **Scalable, Sparse-Masked Sobolev Training.** *A novel sparse-masked Sobolev loss that supervises a selected subset of sensitivities via unbiased vector–Jacobian products—avoiding Hessians,*

*mitigating gradient interference, and scaling stably on GPU to large constrained problems while preserving single-pass inference.* A sparsity-masked, first-order Sobolev loss enforces agreement between a proxy's Jacobian and solver sensitivities on only a carefully chosen subset of partial derivatives. The mask (i) cuts Jacobian memory requirements even further than the vector–Jacobian contraction of Czarnecki et al. (2017), (ii) eliminates the gradient interference that appears under dense supervision, and (iii) stabilizes GPU training on general large-scale optimization problems.

2. **Uniform value–gradient error bounds.** *New uniform value-and-gradient approximation guarantees that extend existing error bounds and supports usage of Sobolev training for constrained* $\arg\min$ *maps.* The paper derives multiple approximation bounds, including a bound that is proportional to the *square* of the training-set covering radius. The bounds extend classical results on value-only matching to joint value-and-gradient consistency, under mild Lipschitz continuity of the exact solution map and $C^2$ neural activations of bounded curvature. This guarantee applies to convex problems and non-convex programs satisfying linear independence constraint qualification (LICQ), second order sufficient conditions (SOSC), and related regularity conditions.

3. **Practical data-generation and implementation.** *An end-to-end pipeline that generates diverse instances and exact KKT sensitivities, applies a task-aware sparsity mask, and trains multi-thousand-variable proxies efficiently on a single GPU, with reproducibility details.* The paper proposes an end-to-end pipeline that automatically (i) produces a broad spectrum of problem instances that cover both nominal and edge-case operating conditions, (ii) retrieves exact parameter sensitivities by differentiating the solver's Karush-Kuhn-Tucker (KKT) system, (iii) sparsifies the resulting Jacobians with a task-aware mask to curb memory and compute demands without sacrificing fidelity, and (iv) relies on modern differentiable-programming tooling—implemented in Julia but agnostic to language—to train multi-thousand-variable proxies comfortably on a single GPU.

4. **Comprehensive empirical validation.** *At modest offline training overhead and unchanged inference cost, empirical results across AC-OPF studies show large reductions in worst-case infeasibility and mean–variance portfolio case supports an absolute dominating mixture-of-experts strategy.* The paper discusses the applicability of the method for training optimization surrogates in different contexts. First, on three industry AC-OPF benchmarks – IEEE-300, PEGASE-1k, and RTE-6k Babaeinejadsarookolaee et al. (2019b) –, it shows that the Sobolev-trained proxy lowers mean-squared error by as much as 56 % and reduces median worst-case constraint violation by up to a factor of four, while keeping the relative optimality gap below 0.22 %. Second, on a self-supervised study on mean–variance portfolio selection, it shows a different pattern: in the tight-risk regime ($\sigma_{\max} \leq 0.10\,\mathcal{B}$), the benchmark that ignores derivatives performs *slightly* better, yet, for looser risk budgets, the Sobolev variant cuts the average optimality gap almost in half (from $18.9 \pm 20.2\%$ to $8.7 \pm 9.5\%$). The pronounced and consistent split in both training and test suggests a mixture-of-experts strategy that calls the benchmark inside the high-constraint region and the Sobolev proxy elsewhere.

Together, the theoretical guarantee and the two contrasting application domains indicate that Sobolev-based training can endow optimization proxies with dependable accuracy *and* high-fidelity gradients, advancing their suitability for safety-critical, large-scale decision systems.

The remainder of the paper is organized as follows. Section 2 reviews the sensitivity-analysis literature that underpins the proposed approach and situates Sobolev training within related work on differentiable optimization layers. Section 3 formalizes the masked Sobolev loss, details the data-generation pipeline for solver sensitivities, and discusses practical implementation choices. Section 4 establishes uniform value-and-gradient error bounds and specifies the conditions under which they hold. Section 5 reports supervised results on three large AC-OPF benchmarks, while Section 6 presents a self-supervised study on mean–variance portfolio selection and motivates a mixture-of-experts extension. Section 7 summarizes observed limitations and open challenges, and Section 8 concludes with avenues for future research.

## 2 BACKGROUND AND LITERATURE REVIEW

Early investigations into the differentiability of solutions to constrained optimization problems can be traced back to Fiacco (1976), which established conditions under which a KKT system exhibits stable, differentiable solutions. In particular, that work emphasized the importance of having a unique primal solution mapping, a unique dual solution, and local stability of active constraints. Under assumptions

such as SOSC, LICQ, and strict complementarity slackness (SCS), these KKT conditions form a set of smooth equations involving both decision variables and parameters, allowing the application of the implicit function theorem to determine how optimal solutions vary with respect to parameter changes.

Subsequent efforts relaxed some of these strict requirements to address degeneracies. For instance, Kojima (1980) developed concepts like Mangasarian-Fromovitz Constraint Qualification (MFCQ) and Generalized Strong Second-Order Sufficient Condition (GSSOSC), which help preserve stability without imposing strict complementarity. This line of research also includes the works of Jittorntrum (1984) and Shapiro (1985), who explored the notion of directional differentiability in the absence of LICQ, and Ralph & Dempe (1995), which provided practical methods for evaluating directional derivatives. Over time, sensitivity analysis expanded to more complex frameworks such as variational inequalities, bi-level formulations Dempe (2002), stochastic programs Shapiro (1990; 1991), and model predictive control Zavala & Biegler (2009); Jäschke et al. (2014), culminating in the creation of robust software implementations Pirnay et al. (2012); Andersson et al. (2019).

Meanwhile, the domain of gradient-based optimization and differentiable programming gained substantial momentum in machine learning Huangfu & Hall (2018); Shin et al. (2023); Lubin et al. (2023); Innes et al. (2019). A significant outcome of this trend is the integration of constrained optimization methods directly into neural networks, which enables end-to-end training pipelines that merge data-driven layers with principled decision models Amos & Kolter (2017); Gould et al. (2022). Several toolkits emerged to streamline these capabilities, including CVXPY-Layers Agrawal et al. (2019), sIPOPT Pirnay et al. (2012), CasADi Andersson et al. (2019), and Theseus Pineda et al. (2022), all of which provide interfaces for treating solvers as differentiable modules. In the Julia ecosystem, `DiffOpt.jl` (Besançon et al. (2024); Rosemberg et al. (2025)) has extended `JuMP.jl` to offer solution sensitivities for convex and non-convex models.

### 2.1 Sensitivity Calculation for Parametric Problems

Let $\mathbf{s}^* = \big(x(p), \lambda(p)\big)$ be a local solution consisting of primal and dual (Lagrange multiplier) variables that satisfy the KKT conditions: $F\big(\mathbf{s}^*, p\big) = 0$, where $F$ encapsulates feasibility, stationarity, and complementary slackness in a single system of equations. Under appropriate regularity assumptions (e.g., LICQ, SOSC, strict complementarity) ensuring a unique local optimum and non-singular derivative $\nabla_{\mathbf{s}} F(\mathbf{s}^*, p)$, one may apply the implicit function theorem to obtain:

$$\nabla_p \mathbf{s}^* = -\Big(\nabla_{\mathbf{s}} F(\mathbf{s}^*, p)\Big)^{-1} \nabla_p F(\mathbf{s}^*, p). \tag{1}$$

This expression reveals how small changes in the parameter vector $p$ propagate through the KKT system to induce changes in both $x(p)$ and its dual variables $\lambda(p)$. In practice, computing these derivatives typically involves linear algebra on the Jacobian of the KKT system. The sensitivity information is valuable for a variety of tasks, including stability analysis, local robustness assessments, and (as studied in this work) using partial derivatives to guide the training of models that approximate the solution mapping $p \mapsto x^*(p)$.

## 3 Methodology

**Sobolev Training.** Sobolev Training augments conventional regression by additionally matching prescribed sensitivities. As sketched in the *upper half of Fig. 1*, given tuples $(p_i, x^* = g(p_i), Dg(p_i) = \partial x^\star / \partial p_i)$, a first-order (masked) Sobolev Loss is defined as

$$\mathcal{L}(\theta) = \frac{1}{N} \sum_{i=1}^N \ell\big(\hat{g}_\theta(p_i), g(p_i)\big) + \frac{\lambda}{N} \sum_{i=1}^N \ell_d\big(\mathcal{M} \odot D\hat{g}_\theta(p_i), \mathcal{M} \odot Dg(p_i)\big), \tag{2}$$

where $D\hat{g}_\theta(p_i)$ denotes the Jacobian of the network at $p_i$ - provided by the same Automatic Differentiation framework (AD) already used for getting gradients of the loss w.r.t. network inputs; $\ell$ and $\ell_d$ measure discrepancies on outputs and directional derivatives respectively, $\lambda$ balances the two terms, and $\mathcal{M} \in \{0, 1\}^{d \times d}$ is a binary mask that through the element wise operation, $\odot$, selects the subset of sensitivities under consideration.

**Sparse Masking.** The masking proposed in this paper helps mitigate the memory cost of full-Jacobian matching and alleviates gradient conflicts reported in large-scale constrained learning Liu et al. (2024). Empirical results show that extreme sparsity—masking between 75 % and 95 % of

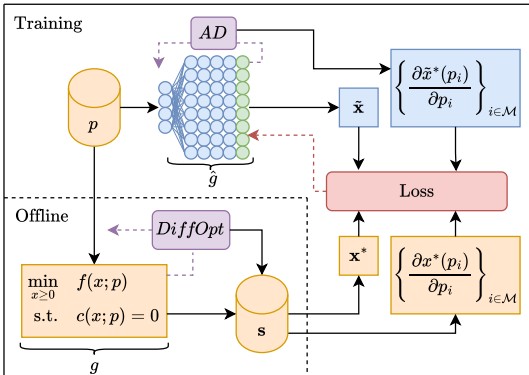

Figure 1: End-to-end pipeline for Sobolev training of optimization proxies. The upper half shows the online training loop driven by automatic differentiation (AD); the lower half is the offline oracle that supplies ground-truth solutions $x^\star$ and (masked) sensitivities via a Differentiable Optimization (`DiffOpt.jl`) layer. Solid arrows are forward evaluations, dashed arrows are represent Jacobian/Gradient calculations (departing from outputs and reaching inputs).

Jacobian entries (i.e. keeping only 5–25 %)—yields the most stable training and lowest test-set MSE on large surrogate models. An ablative study on the IEEE-300 case instance for AC-OPF, detailing how performance varies with mask density, is provided in Appendix F.

**Supervised vs Self-Supervised**   The expression $\ell$ can act as a purely supervised regression loss when ground-truth solutions and their derivatives are available, exploiting the *optimization model itself* as a teacher. In situations where exact solutions are unknown, expensive or may represent a prohibitive hard mapping, the optimization objective regularized by penalties for constraint violation (also know as the Lagrangian function) provides an unsupervised signal: $\tilde{f}(x; p) := f(x; p) + \beta \|c(x; p)\|_2^2 + \gamma \langle x \rangle_-$, with $\beta, \gamma > 0$ and $\langle \cdot \rangle_-$ denoting the parts of the $x$ not respecting the non-negativity constraint.

**Feasibility-Preserving Proxies**   While the training of optimization proxies also aims at reducing infeasibility, the raw output predicted by the neural network, $\hat{g}_\theta : p \to \tilde{x}$, is not required to satisfy the problem constraints: $\mathcal{C}(p) = \{ x \in \mathbb{R}^n \mid c(x; p) = 0, \ x \geq 0 \}$. In situations where a projection, $proj_{\mathcal{C}(\mathbf{p})}\tilde{\mathbf{x}}$, is easily computed and has (at least) directional derivatives, *feasibility-enforcing layers* (e.g. last network layer in Fig. 1) have been shown to improve performance. This is the case for one of the applications in the present study presented in Section 6. Accordingly, feasibility projections and Sobolev training are complementary rather than competing; and for general constraint sets without cheap projections, Sobolev supervision offers a significantly more efficient way to improve feasibility.

**Dataset Creation.**   Constructing a dataset that faithfully represents the solver's behavior across both feasible and infeasible regions requires a blend of sampling strategies. Uniform "box" sampling draws within believed parameter feasible bounds, while incremental "line" excursions along single coordinates expose the boundary limits of feasibility. Additionally, realistic distributions over parameters, can be used when they are known (as in the case of Power Systems). Combining these three procedures in fixed proportions yields a dataset that spans the operating domain without excessive redundancy.

In Sobolev training, where a supervised regression loss is needed for at least matching sensitivities, the workflow proceeds as follows. An optimization instance is solved for every parameter realization $p_i$ in the training set with an appropriate solver such as IPOPT Biegler & Zavala (2009) (Optimization block in Fig. 1) to obtain the primal (sometimes local) optimum $x^\star(p_i)$. Solver sensitivities $Dg(p_i)$ are extracted through sensitivity analysis using a software such as `DiffOpt.jl`, and finally a sparse mask $\mathcal{M}$ is randomly sampled for each instance. Once the dataset is created [1], the Sobolev training

---

[1]If one has already stored the primal and dual solutions and the problem structure is available, there is no need to re-solve to compute derivatives.

loop proceeds much like standard training. The *lower half of Fig. 1* sketches the order of the different procedures in the offline dataset creation and how it feeds into training.

# 4 THEORETICAL PERSPECTIVE

Many classical universal approximation theorems show that neural networks can approximate continuous functions arbitrarily well in $L^2-$ or uniform norms. However, matching derivatives requires approximation in a Sobolev norm. Earlier results Czarnecki et al. (2017) show that standard networks with sufficiently rich activation functions can also approximate derivatives.

One key advantage of incorporating derivatives is the reduction in sample complexity. In certain families of functions, it can take fewer input-output-derivative samples to identify a target function than input-output pairs alone. Indeed, the training set conveys, not only the function value at a point, but also how it changes locally, reducing the degrees of freedom in the hypothesis space. As in classical polynomial-fitting arguments, knowledge of derivatives at a point can disambiguate many potential fits. Recent results have demonstrated how approximating derivatives (either explicitly or implicitly) Zhang et al. (2022); Rosemberg et al. (2024) can help achieve small generalization errors in the test set. This effect becomes especially significant in high-dimensional problems or in the presence of strong non-linearities, where matching values alone might fail to capture the local geometry of $f$. Moreover, classical results from sensitivity analysis (e.g., Liu (1995)) guarantee that for a large class of convex and well-behaved nonconvex optimization problems, the optimal solution mapping is continuous or even Lipschitz-continuous. In particular, if the optimization problem is convex and satisfies regularity conditions such as Slater's condition, then small perturbations in the problem parameters lead only to bounded—and typically linear—changes in the optimal solution. Consequently, the derivatives (when they exist) are finite and well-behaved, ensuring that a Sobolev-trained model which matches both the values and sensitivities at training points can interpolate accurately between them. This bounded, regular behavior of the solution mapping is precisely the scenario where Sobolev Training excels, as it can exploit these local regularity properties to greatly reduce sample complexity and improve generalization.

The ensuing three approximation theorems formalize this intuition, bounding the proxy's uniform error under value-only, Jacobian-only, and joint Sobolev matching.

Let $\mathcal{P} \subset \mathbb{R}^d$ be a compact parameter space and $\mathcal{T} = \{p_i\}_{i=1}^N \subset \mathcal{P}$ a finite training set whose *covering radius* is

$$\delta := \sup_{p \in \mathcal{P}} \min_{1 \le i \le N} \|p - p_i\|.$$

Denote the exact solution operator by $g : \mathcal{P} \to \mathbb{R}^n$ and the learned proxy by $\hat{g}_\theta : \mathcal{P} \to \mathbb{R}^n$. Sobolev training enforces

$$\hat{g}_\theta(p_i) = g(p_i), \qquad D\hat{g}_\theta(p_i) = Dg(p_i), \qquad \forall\, p_i \in \mathcal{T}.$$

**Assumption 1 (Value-Lipschitz)** *There exist inherent values $L_g, L_{\hat{g}} > 0$ such that $\|g(p) - g(q)\| \le L_g \|p - q\|$ and $\|\hat{g}_\theta(p) - \hat{g}_\theta(q)\| \le L_{\hat{g}} \|p - q\|$ for all $p, q \in \mathcal{P}$.*

**Assumption 2 (Derivative-Lipschitz)** *There exist $M_g, M_{\hat{g}} > 0$ with $g, \hat{g}_\theta \in C^1(\mathcal{P})$ and $\|Dg(p) - Dg(q)\| \le M_g \|p - q\|$, $\|D\hat{g}_\theta(p) - D\hat{g}_\theta(q)\| \le M_{\hat{g}} \|p - q\|$ for all $p, q \in \mathcal{P}$.*

**Theorem 1 (Value Matching Only)** *Let $\delta$ be the* covering radius *of $\mathcal{P}$. Under Assumption 1,*

$$\sup_{p \in \mathcal{P}} \|\hat{g}_\theta(p) - g(p)\| \le (L_g + L_{\hat{g}})\, \delta.$$

**Proof 1** *Fix any $p \in \mathcal{P}$ and choose $p_j \in \mathcal{T}$ with $\|p - p_j\| \le \delta$. By the triangle inequality and Lipschitz bounds, $\|\hat{g}_\theta(p) - g(p)\| \le$*

$$\|\hat{g}_\theta(p) - \hat{g}_\theta(p_j)\| + \|g(p_j) - g(p)\| \le L_{\hat{g}} \|p - p_j\| + L_g \|p - p_j\| \le (L_g + L_{\hat{g}})\delta.$$

*Taking the supremum over $p$ completes the proof.*

**Theorem 2 (Jacobian Matching Only)** *Let $\delta$ be the* covering radius *of $\mathcal{P}$. Under Assumption 2,*

$$\sup_{p \in \mathcal{P}} \|D\hat{g}_\theta(p) - Dg(p)\| \le (M_g + M_{\hat{g}})\, \delta.$$

**Proof 2** *Repeat the argument above with $Dg$ and $D\hat{g}_\theta$ in place of $g$ and $\hat{g}_\theta$ and with $M$-constants instead of $L$-constants.*

**Theorem 3 (Sobolev Guarantee)** *Let $\delta$ be the covering radius of $\mathcal{P}$. Under Assumptions 2 and with both value and Jacobian interpolation on $\mathcal{T}$,*

$$\sup_{p \in \mathcal{P}} \left\| \hat{g}_\theta(p) - g(p) \right\| \leq \tfrac{1}{2} \left( M_g + M_{\hat{g}} \right) \delta^2.$$

**Proof 3** *Let $p \in \mathcal{P}$ be arbitrary and pick its nearest training point $p_j$ so that $h := p - p_j$ satisfies $\|h\| \leq \delta$. For any $C^1$ map $f$ the fundamental theorem of calculus gives*

$$f(p) = f(p_j) + Df(p_j)h + \int_0^1 \big( Df(p_j + th) - Df(p_j) \big) h \, dt.$$

*Define the remainder $R_f(h) := \int_0^1 \big( Df(p_j + th) - Df(p_j) \big) h \, dt$. Derivative-Lipschitz continuity yields $\|R_f(h)\| \leq \frac{1}{2} M_f \|h\|^2$. Applying this to $f = g$ and $f = \hat{g}_\theta$ and using exact interpolation at $p_j$ cancels the zeroth- and first-order terms, so*

$$\hat{g}_\theta(p) - g(p) = R_{\hat{g}_\theta}(h) - R_g(h) \quad \implies \quad \|\hat{g}_\theta(p) - g(p)\| \leq \tfrac{1}{2}(M_g + M_{\hat{g}}) \|h\|^2 \leq \tfrac{1}{2}(M_g + M_{\hat{g}}) \, \delta^2.$$

**Remark 1** *Compactness of $\mathcal{P}$ plus standard regularity conditions (LICQ, SOSC, strict complementarity) guarantee $g \in C^{1,1}(\mathcal{P})$ with bounded Jacobian, so the assumptions are generically met. Feed-forward networks whose activations are $C^2$ with bounded curvature (e.g. $\tanh$, softplus) satisfy the same properties on compact domains, hence Theorems 1–3 apply directly to the proxies considered in this paper.*

## 5 SUPERVISED LEARNING APPLICATION: OPTIMAL POWER FLOW

Power-system optimization has surged in importance due to network expansion, deep renewable-energy penetration, and sustainability targets Soares et al. (2022); Pozo et al. (2012); Pozo & Contreras (2012); Barry et al. (2022). A central computational task, the AC Optimal Power Flow (AC-OPF) problem, minimizes total generation cost via the objective (3a) while satisfying Kirchhoff's current-law balance at every bus (3b), Ohm's-law branch relations (3c)–(3d), voltage-magnitude limits (3e), generator operating bounds (3f), and thermal capacity constraints on apparent power flows (3g). High-accuracy AC-OPF solutions enable real-time risk-aware market clearing Tam (2011); Chen (2023), day-ahead security-constrained unit commitment Sun et al. (2017), transmission switching optimization Fisher et al. (2008), and long-term expansion planning Verma et al. (2016). The concise formulation in Model 3 employs complex voltages, injections, demands, and branch flows, omitting detailed bus-shunt and transformer models for brevity, and follows the reference implementation in `PowerModels.jl` Coffrin et al. (2018). [2] The reference model notation is detailed in Appendix A.

$$\min_{S^{\mathrm{g}}, S^{\mathrm{f}}, \mathbf{V}} \quad \sum_{i \in \mathcal{N}} c_i(S_i^{\mathrm{g}}) \tag{3a}$$

$$\text{s.t.} \quad S_i^{\mathrm{g}} - \boxed{S_i^{\mathrm{d}}} = \sum_{(i,j) \in \mathcal{E}_i \cup \mathcal{E}_i^R} S_{ij}^{\mathrm{f}} \qquad \forall i \in \mathcal{N} \tag{3b}$$

$$S_{ij}^{\mathrm{f}} = (Y_{ij} + Y_{ij}^c)^\star \mathbf{V}_i \mathbf{V}_i^\star - Y_{ij}^\star \mathbf{V}_i \mathbf{V}_j^\star \qquad \forall (i,j) \in \mathcal{E} \tag{3c}$$

$$S_{ji}^{\mathrm{f}} = (Y_{ij} + Y_{ji}^c)^\star \mathbf{V}_j \mathbf{V}_j^\star - Y_{ij}^\star \mathbf{V}_i^\star \mathbf{V}_j \qquad \forall (i,j) \in \mathcal{E} \tag{3d}$$

$$\underline{\mathbf{v}_i} \leq |\mathbf{V}_i| \leq \overline{\mathbf{v}_i} \qquad \forall i \in \mathcal{N} \tag{3e}$$

$$\underline{S_i^{\mathrm{g}}} \leq S_i^{\mathrm{g}} \leq \overline{S_i^{\mathrm{g}}} \qquad \forall i \in \mathcal{N} \tag{3f}$$

$$|S_{ij}^{\mathrm{f}}|, |S_{ji}^{\mathrm{f}}| \leq \overline{S_{ij}}^2 \qquad \forall (i,j) \in \mathcal{E} \tag{3g}$$

The nonconvexity and nonlinear physics in AC-OPF lead to poor scaling of solution time as network size and scenario count grow, often precluding its direct use in large-scale or highly uncertain settings

---

[2]Problem parameters emphasized in Red Box.

O'Neill et al. (2011). However, the need to solve many similar OPF instances has spurred research on learning a parametric surrogate $p \mapsto x^*(p)$ that delivers near-optimal solutions in a single forward pass, thereby bypassing expensive iterative algorithms Park & Van Hentenryck (2023); Chatzos et al. (2020) for each parameter instance $p = S^{\mathrm{d}}$. One can train a neural network to predict $x^*(p)$ from $p$ using the standard MSE or similar loss, but Sobolev training augments this with a derivative penalty that aligns $\frac{\partial \hat{x}_\theta}{\partial p}$ with $\frac{\partial x^*(p)}{\partial p}$.

## 5.1 EXPERIMENTAL SETTING

Two neural-network proxy models were evaluated on the AC-OPF problem in Model 3: one trained using a standard MSE loss (hereafter refereed to as Benchmark/Bench), and another trained with an additional Sobolev loss term that leverages derivative information (hereafter refereed to as Sobolev). Both models employed the same fully connected architecture with a single hidden layer of width 320, trained over approximately $10K$ samples for $20K$ epochs. The validation and test set comprised of around $5K$ problem instances each. The evaluation uses three test cases from PGLib Babaeinejadsa-rookolaee et al. (2019a) with up to 6468 buses. Appendix B provides, for each system, case details (e.g., number of buses and branches) and Implementation aspects.

## 5.2 METRICS

Three metrics are employed to assess both solution accuracy and feasibility. Denote $\tilde{x}$ the proxy's prediction, $x^*$ the true solution, $\tilde{z} = f(\tilde{x})$, and $z^* = f(x^*)$. 1) Mean Squared Error: $\mathrm{MSE} = \frac{1}{n}\|\tilde{x} - x^*\|_2^2$; 2) Optimality Gap: $\mathrm{GAP} = \frac{|\tilde{z} - z^*|}{|z^*|}$, and 3) Absolute Infeasibility (INF):

$$\mathrm{INF} = \frac{1}{|\mathcal{E}| + |\mathcal{I}|} \sum_{\substack{c \in \mathcal{E} \\ g \in \mathcal{I}}} \begin{cases} |\,c(\hat{x})\,|, & c(x) = 0 \text{ (equality)}, \\ \max\{\,g(\hat{x}), 0\}, & g(x) \le 0 \text{ (inequality)}. \end{cases}$$

## 5.3 EXPERIMENTAL RESULTS

Table 1: Average performance of OPF Proxy Models (Sobolev vs MSE).

| Case | Metric | Sobolev | Bench |
|------|--------|---------|-------|
| ieee300 | MSE | **0.0065** | 0.0070 |
| ieee300 | GAP | 0.22% | **0.02%** |
| ieee300 | INF | **0.0099** | 0.0116 |
| pegase1k | MSE | **0.0072** | 0.0089 |
| pegase1k | GAP | 0.15% | **0.07%** |
| pegase1k | INF | **0.0066** | 0.0098 |
| rte6k | MSE | **0.0004** | 0.0009 |
| rte6k | GAP | **0.07%** | **0.07%** |
| rte6k | INF | **0.0042** | 0.0059 |

Table 1 reports the average performance of both models on the held-out test sets. The Sobolev-trained proxy achieves lower MSE and average absolute infeasibility across all three network cases. Not surprisingly since it has fewer degrees of freedom, Sobolev incurs a slightly higher optimality gap on IEEE-300 and PEGASE-1k (0.22% vs. 0.02%, and 0.15% vs. 0.07%, respectively); on RTE-6k both models match at 0.07%.

Moreover, the frequency with which higher constraint violations occurred across test instances was further examined using the relative reduction in maximum infeasibility (RMI) under Sobolev training, defined for instance $i$ as

$$\mathrm{RMI_i} = 100\% \times \frac{\max_j(\mathrm{Infeas}_{\mathrm{MSE}_{i,j}}) - \max_j(\mathrm{Infeas}_{\mathrm{Sobolev}_{i,j}})}{\max_{i,j}(\mathrm{Infeas}_{\mathrm{Sobolev}_{i,j}})}.$$

In this definition, $\mathrm{Infeas}_{\mathrm{MSE},i,j}$ and $\mathrm{Infeas}_{\mathrm{Sobolev},i,j}$ are the absolute violations of constraint $j$ on instance $i$ for the MSE-only and Sobolev-trained models, respectively. The numerator measures how

much the worst-case violation on instance $i$ is reduced by Sobolev training, and the denominator normalizes by the single largest violation observed across all test instances and constraints (to avoid instances with zero violation). Hence, a positive $\text{RMI}_i$ indicates the percentage by which Sobolev loss decreases the maximum infeasibility relative to the MSE benchmark.

The Sobolev-based model rarely yields worse violations than the MSE-only benchmark (fewer than 15% of cases). Median improvements in maximum infeasibility are approximately 18%, 400%, and 180% for IEEE-300, PEGASE-1k, and RTE-6k, respectively; moreover, Sobolev training substantially suppresses the extreme outliers observed with MSE loss. Around 90% of the remaining infeasibility originates from minor power-flow mismatches (3b–3d), suggesting that coupling with feasibility-focused losses or projection techniques could yield further gains. Figure 3 in Appendix D presents these results as violin plots of the RMI.

Overall, while Sobolev loss incurs a small trade-off in optimality gap (e.g., increasing from 0.02% to 0.22% in IEEE-300 and from 0.07% to 0.15% in PEGASE-1k), the dramatic reduction in average and worst-case constraint violations makes the Sobolev-trained proxy significantly more reliable for safety-critical OPF applications.

# 6 (Semi) Self-Supervised Case Study: Mean–Variance Portfolio Optimization

Self-Supervised Learning (SSL) of optimization proxies has great benefits since it only requires a distribution of the input data and avoids the need for optimal solutions. Moreover, on some applications, SSL becomes a necessity as supervised learning may fail to outperform even basic strategies. This happens for the classical Markowitz portfolio-selection (MPS) problem which seeks a weight vector $x \in \mathbb{R}^n$ for $n$ tradable assets that maximizes expected return subject to a risk budget:

$$\max_{x \succeq 0} \quad \boxed{\mu}^\top x \quad \text{s.t.} \quad x^\top \Sigma x \leq \boxed{\sigma_{\max}^2}, \quad \mathbf{1}^\top x \leq \mathcal{B},$$

Here $\mu$ denotes expected returns, $\Sigma$ the covariance matrix, and $\mathcal{B}$ the available capital. The quadratic risk constraint can be rewritten as the second-order cone $\|\Sigma^{1/2}x\|_2 \leq \sigma_{\max}$, where $\Sigma^{1/2}$ is any Cholesky factor and $\sigma_{\max}$ limits portfolio standard deviation. The problem is convex, ensuring global optimality, yet solving it to completion may be infeasible in sub-second trading loops when $n$ is large.

**Motivation for a proxy.** Practitioners often impose tight risk budgets in mean–variance models as a pragmatic hedge against *distributional ambiguity* in returns since multiple data-consistent distributions can explain the same history. In principle, this ambiguity is best handled with distributionally robust optimization (e.g., Van Parys et al. (2021)), but context-conditional DRO often reduces to large complex programs with unfavourable scaling, making it impractical for high-frequency rebalancing. As a practical alternative to a fixed, tight risk budget, the above-described version of the mean-variance portfolio proposes context-adaptive risk constraints. Covariances are still estimated over long horizons (as done in the usual setting) and updated infrequently. In low-latency settings $\Sigma$ can therefore be considered fixed, and the optimization treated as parameterized by the rapidly evolving signals $(\mu, \sigma_{\max})$. A neural proxy that instantaneously maps these signals to an approximate solution $x^\star(\mu, \sigma_{\max})$ enables portfolio re-balancing at the pace required for high-frequency trading.

**Training strategy.** Supervised learning for MPS resulted in poor proxies with over 95% optimality gaps, as detailed in Appendix E. The results in this section reports on the semi SSL approach. Feasibility can be enforced through inexpensive projection steps Qiu et al. (2024). Regularity conditions needed for reliable solver sensitivities hold, making the task suitable for *Sobolev training*, which augments a value-matching loss with a Jacobian term. A practical complication is sparsity: typically fewer than 15 % of assets receive allocations above 0.01 % of budget, creating a difficult mapping for purely supervised learners. Two proxy networks equipped with a feasibility layer are therefore compared: **Benchmark** which is trained only on objective values; and **Sobolev** which is trained on values *and* solver sensitivities. For the Sobolev approach, after roughly half of the maximum number of training epochs (20K)—the time needed to solve all instances in the dataset—training transitioned from value only learning to using the labeled optimal solutions to compute sensitivities. Data generation is detailed in Appendix C.

Table 2: Test-set gap (mean $\pm$ std)

| $\sigma_{\max}$ | Sobolev | Bench |
|---|---|---|
| $\leq 10\%$ | $29.5 \pm 26.9$ | $14.7 \pm 21.2$ |
| $> 10\%$ | $8.7 \pm 9.5$ | $18.9 \pm 20.2$ |


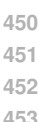
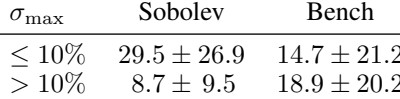

Figure 2: Change in Relative Optimality Gap ($\text{Gap}_{\text{Sobolev}} - \text{Gap}_{\text{Benchmark}}$). Negative Values Indicate Improvement.

**Results.** Figure 2 displays the gap difference across train and test instances. In the tight-risk regime $\sigma_{\max} \leq 0.10\,\mathcal{B}$, where first-order information is less informative, the Sobolev proxy under-performs the benchmark. Outside that region, Sobolev training achieves a clear advantage. Table 2 provides detailed statistics; the pronounced split suggests the benefit of a mixture-of-experts approach that calls the benchmark in the highly constrained region and the Sobolev proxy elsewhere.

## 7 LIMITATIONS

**Nested Automatic Differentiation and Jacobian Overconstraint.** Initial experiments encountered two critical challenges: most neural–network frameworks did not efficiently support higher-order or derivative-based losses on GPUs, leading to training times of up to 8 hours for 6,000 epochs, and enforcing every entry of the solver Jacobian in each batch overconstrained the model, degrading performance relative to the MSE-only benchmark. To address these issues, nested-AD capabilities of `Lux.jl` were employed, dramatically shortening training times and enabling live monitoring of loss components. Additionally, random masking of 95% of the derivative entries in each mini-batch alleviated the overconstraint, resulting in faster convergence and improved predictive accuracy. Across our settings, the increases epoch time from the sparse derivative matching was approximately $1.2\times$–$2.6\times$ compared to the benchmark (as shown in Appendix Table 5). Nevertheless, this cost is paid only at training time, while inference remains a single forward pass (i.e., no Jacobian computation).

**Regularity Conditions and Sensitivity Smoothness.** The theoretical guarantees of Sobolev Training (see Sec. 2 and 4) rely on regularity conditions—such as LICQ, SOSC, and strict complementarity—to ensure a smooth, well-defined mapping from parameters to solutions. Although many OPF instances satisfy these assumptions, occasional degeneracies (e.g., non-unique active sets) can produce invalid or discontinuous sensitivities. Techniques such as the "fix-and-relax" method (Pirnay et al., 2012) or the corrective strategies in (Andersson et al., 2019) can restore valid derivative information. In the conducted experiments, such irregular cases were present but did not materially affect overall proxy performance.

**Uninformative Sensitivities.** Certain problem classes—especially those with stepwise or piecewise-constant mappings from parameters to variables—naturally yield zero or uninformative sensitivities. A prime example is the Lagrangian dual variables in linear programming constraints, whose gradients vanish across broad parameter intervals. Mitigation strategies may include regularization methods or alternative loss formulations designed to handle sparse or zero-derivative signals (Jungel et al., 2023).

**Semi vs Fully Self-Supervised** In the MPS study, the self-supervised variant substantially outperforms supervised baselines, which is the reason for its choice even though the sensitivities were computed from true labels. Accordingly, using labels solely to form sensitivity targets does not contradict the premise; it is a pragmatic, performance-oriented choice that preserves the benefits of self-supervision. Nevertheless, for industry-scale applications, ground-truth labels can be prohibitively expensive to obtain, which would motivate a **fully** self-supervised training. In turn, this would require that the implicit function differentiation be computed in **GPU** and in **batch** from the proxy's predictions. Although recent progress in GPU batch-compatible linear-system solvers (e.g., cuDSS) is promising, further improvements in numerical stability and precision are needed to ensure reliable performance.

## 8 Conclusion

This work demonstrates how Sobolev Training can serve as an effective end-to-end optimization proxy by embedding solver-based sensitivities into the learning process. A theoretical framework under Lipschitz continuity provides performance guarantees, showing that matching both values and derivatives controls the approximation error in terms of training set density. Empirical evaluation on three large AC-OPF systems shows that Sobolev training yields lower prediction error, tighter constraint satisfaction, and *far fewer extreme infeasibilities* than an MSE-only proxy; moreover, on a (semi) self-supervised mean–variance portfolio task it *slashes the average optimality gap by roughly 50 % in the medium-risk regime* while matching the baseline in tighter-risk settings.

Future research may explore the incorporation of second-order derivative information to further improve approximation fidelity, adapt the Sobolev approach to mixed-integer and stochastic optimization problems, and develop feasibility-restoration or projection techniques to handle instances where solver regularity conditions are marginally violated. These extensions promise to broaden the applicability and robustness of optimization proxies in large-scale industrial settings.

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

# A  AC-OPF NOMENCLATURE

This appendix lists every symbol and constraint that appears in the AC optimal power-flow formulation of Model 3.

## A.1  SETS AND INDICES

| Symbol | Definition |
|---|---|
| $\mathcal{N}$ | Set of buses (nodes) in the network, indexed by $i, j$ |
| $\mathcal{E}$ | Set of directed branches, indexed by $(i, j)$ |
| $\mathcal{E}_i$ | Branches whose *from-bus* is $i$ |
| $\mathcal{E}_i^R$ | Branches whose *to-bus* is $i$ |

## A.2  DECISION VARIABLES

| Variable | Description | Units |
|---|---|---|
| $S_i^{\mathrm{g}}$ | Complex net power generated at bus $i$ | MVA |
| $\mathbf{V}_i$ | Complex voltage phasor at bus $i$ | p.u. or kV |
| $S_{ij}^{\mathrm{f}}$ | Complex apparent power flow - $i$ to $j$ | MVA |

## A.3  CONSTANTS

| Constant | Description | Units |
|---|---|---|
| $c_i(\cdot)$ | Generation cost curve at bus $i$ | \$ |
| $Y_{ij}$ | Series admittance of branch $(i, j)$ | p.u. |
| $Y_{ij}^c$ | Shunt admittance of branch $(i, j)$ | p.u. |
| $\underline{\mathbf{v}_i}, \overline{\mathbf{v}_i}$ | Voltage magnitude limits at bus $i$ | p.u. or kV |
| $\underline{S_i^{\mathrm{g}}}, \overline{S_i^{\mathrm{g}}}$ | Generator capability limits at bus $i$ | MVA |
| $\overline{S_{ij}}$ | Thermal limit of branch $(i, j)$ | MVA |

## A.4  PARAMETERS

| Parameter | Description | Units |
|---|---|---|
| $S_i^{\mathrm{d}}$ | Forecast demand at bus $i$ | MVA |

## A.5  CONSTRAINT NOTES FOR MODEL 3

C1. (3a)  Minimise total generation cost $\sum_i c_i(S_i^{\mathrm{g}})$.
C2. (3b)  Nodal power balance (Kirchhoff's current law).
C3. (3c)–(3d)  Non-linear AC power-flow relations (Ohm's law).
C4. (3e)  Voltage magnitude limits.
C5. (3f)  Generator capability limits.
C6. (3g)  Branch thermal limits.

LEGEND

Variables: $S_i^{\mathrm{g}}$, $S_{ij}^{\mathrm{f}}$, $\mathbf{V}_i$    Parameters: all other symbols    $|\cdot|$ = magnitude, $^\star$ = complex conjugate

---

[2]Both directions of a physical line are treated explicitly: $(i, j)$ and $(j, i)$ may appear separately in $\mathcal{E}$.

## B  CASE DETAILS & IMPLEMENTATION

Table 3 reports, for each system, the number of buses $|\mathcal{N}|$, branches $|\mathcal{E}|$ and generators $|\mathcal{G}|$, as well as the nominal total demand ($\mathrm{P}_{\mathrm{ref}}^d$) and its range across the dataset ($[\underline{\mathbf{P}}^d, \bar{\mathbf{P}}^d]$).

Table 3: Statistics of the PGLib test cases.

| System | $|\mathcal{N}|$ | $|\mathcal{E}|$ | $|\mathcal{G}|$ | $\mathbf{P}_{\mathrm{ref}}^d$ | $[\underline{\mathbf{P}}^d, \bar{\mathbf{P}}^d]$ |
|---|---|---|---|---|---|
| ieee300 | 300 | 411 | 69 | 263 | [ 210,  280] |
| pegase1k | 1354 | 1991 | 260 | 781 | [ 625,  820] |
| rte6k | 6468 | 9000 | 399 | 1109 | [ 887, 1164] |

All experiments are implemented in the Julia programming language Bezanson et al. (2017), which combines a high-level, dynamic syntax with just-in-time compilation to deliver performance on par with statically compiled languages. Julia's multiple-dispatch paradigm and extensive numerical ecosystem make it particularly well-suited for blending optimization modeling, automatic differentiation, and machine learning in a single environment.

Transmission network and generator data are ingested via `PowerModels.jl`, a domain-specific package that parses PGLib case files and constructs corresponding `JuMP.jl` models. `JuMP.jl` serves as the algebraic modeling layer, enabling concise formulation of AC-OPF constraints and objectives and delegating solution to state-of-the-art nonlinear solvers.

Dataset generation is automated using the external package `LearningToOptimize.jl` (L2O.jl), which samples random parameter vectors, solves each OPF instance, and records both the optimal primal variables and the associated sensitivities. Sensitivities are extracted using `DiffOpt.jl`, an extension of `JuMP.jl` that computes solver-level derivatives via nested automatic differentiation. The resulting triples $(p_i, x^*(p_i), Dx^*(p_i))$ form the training corpus for the Sobolev regression loss.

Neural-network training was split across two frameworks. The benchmark value-only models were implemented in `Flux.jl`, chosen for its mature support and high computational throughput. In contrast, Sobolev training was carried out in `Lux.jl`, a separate Julia deep-learning library inspired by the Flux API (but architecturally independent) with with GPU-compatible nested higher-order derivatives.

Table 4: Training hyper–parameters for each proxy network.

| System | Layers | Activation | Batch size | $\Lambda$ | Mask sparsity (%) |
|---|---|---|---|---|---|
| pegase1k | [320, 320] | ReLU | 32 | 0.14 | 5 |
| ieee300 | [320, 320] | Sigmoid | 32 | 0.30 | 5 |
| rte6k | [320, 320] | ReLU | 32 | 0.1 | 25 |
| MPS | [1024, 1024] | LeakyReLU | 32 | 4.35 | 5 |

**Optimizer.** All models use `Adam` ($\eta = 0.001$, $\beta = (0.9, 0.999)$, $\epsilon = 10^{-8}$).

**Resources.** Experiments are carried out on Intel(R) Xeon(R) Gold 6226 CPU @ 2.70GHz machines with NVIDIA Tesla A100 GPUs on the Phoenix cluster PACE (2017).

Table 5: Average epoch training time (seconds) for benchmark (value-only) and Sobolev training.

| Experiment | Benchmark (s) | Sobolev (s) |
|---|---|---|
| Supervised OPF (`ieee300`) | 0.2892 | 0.6312 |
| Supervised OPF (`pegase1k`) | 0.4369 | 0.9593 |
| Supervised OPF (`rte6k`) | 0.9625 | 1.5551 |
| (Semi) Self-Supervised OPF (`ieee300`) | 0.6396 | 0.7714 |
| Self & Supervised MPS | 0.0941 | 0.2432 |

## C  DATA GENERATION MPS

To evaluate the approach on arbitrary asset universes rather than a fixed pool, we drew mean vectors uniformly from the $\ell_2$ unit ball and generated positive semidefinite covariances via two complementary procedures:

- Random dense construction: Sample $A \in [0,1]^{n \times n}$ i.i.d. and set $\Sigma = A^\top A$.
- Factor-model construction: Simulate returns with Gaussian idiosyncratic terms and a small number of Gaussian common factors. For each 10k-sample history, factor loadings were randomly scaled in $[0, 0.3]$, and each factor influenced a random fraction of assets between $10\%$ and $100\%$. We then computed the empirical covariance $\Sigma$.

In both cases we verified symmetry/Hermiticity and positive definiteness to avoid numerical issues. The factor-model route was used to diversify the range of covariance structures without aiming to calibrate a fully realistic market model; it was not required for spanning PSD matrices but helped enrich the dataset. The portfolio dimension was n = 500.

# D ADDITIONAL RESULTS SUPERVISED AC-OPF

Figure 3 presents violin plots of the relative reduction in maximum infeasibility (RMI) under Sobolev training.

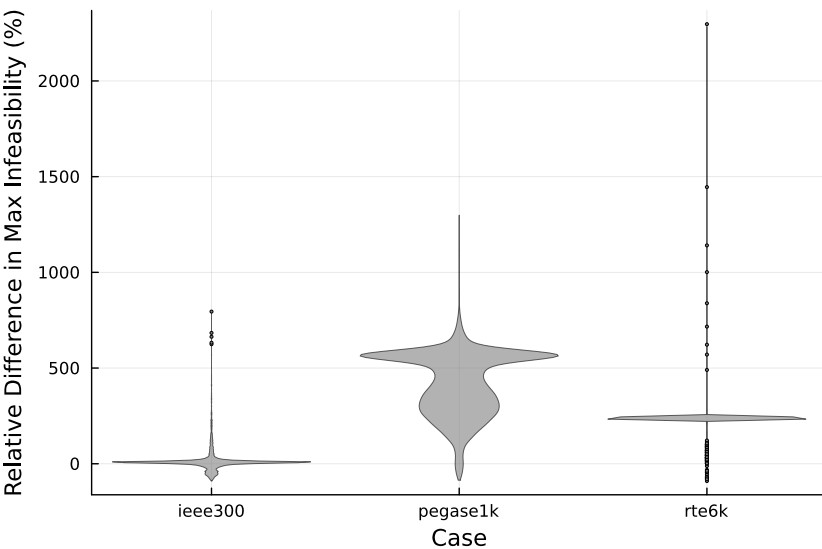

Figure 3: Relative Max Infeasibility Difference (%).

# E SUPERVISED MEAN–VARIANCE PORTFOLIO OPTIMIZATION

This appendix evaluates a fully supervised proxy for the mean–variance task by solving each instance offline to obtain exact portfolio weights and (where available) sensitivities. The results in Table 6 and Figure 4—showing large mean optimality gaps and high variance even with perfect labels—underscore why the main paper relies on a self-supervised training strategy, which avoids these challenges.

| GAP | Sobolev | Bench |
|---------|---------|-------|
| Mean | 92.3 | 93.41 |
| STD | 2.48 | 2.31 |
| Maximum | 98.99 | 99.15 |
| Minimum | 66.33 | 90.14 |

Table 6: Test-set gap

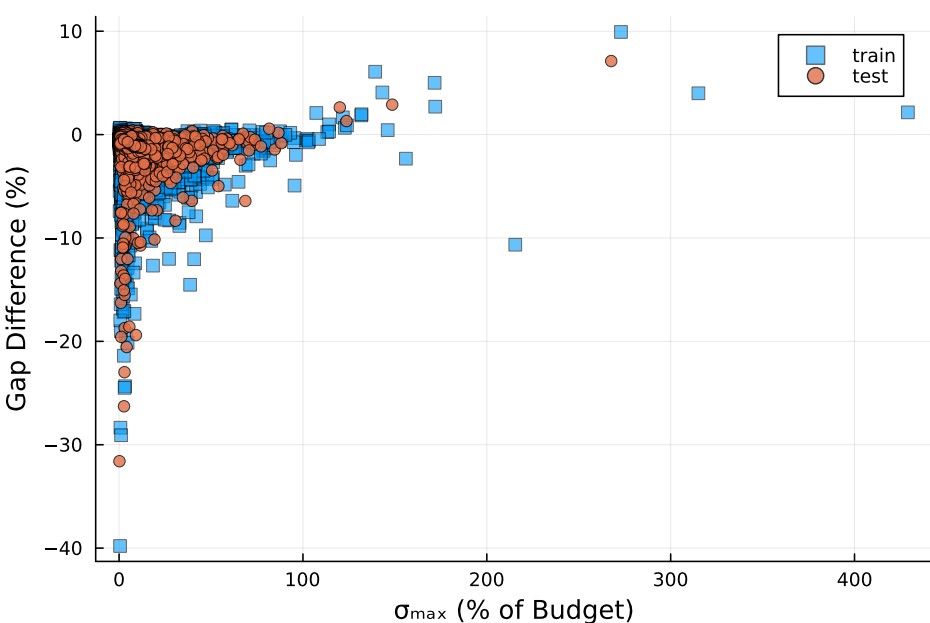

Figure 4: Change in Relative Optimality Gap ($\text{Gap}_{\text{Sobolev}} - \text{Gap}_{\text{Benchmark}}$). Negative Values Indicate Improvement.

# F  MASK IMPACT

As reported in Table 7, extreme sparsity (5–10%) achieves the lowest MSE and INF values, indicating more accurate and feasible solutions under tighter constraints. Increasing the density beyond 25% not only degrades MSE performance but also leads to a sharp rise in INF—particularly for the fully dense (100%) variant, where infeasibility increases more than $5\times$ relative to the 5% case. This trend supports the hypothesis that excessive gradient information introduces contention among constraints, ultimately hampering convergence and solution feasibility.

Table 7: Impact of Jacobian-mask sparsity on test MSE (`ieee300`)

| Mask sparsity (%) | MSE | GAP | INF |
|---|---|---|---|
| 5 | 0.0065 | 0.22% | 0.0099 |
| 10 | 0.0088 | 0.33% | 0.0094 |
| 25 | 0.0087 | 0.29% | 0.0141 |
| 100 | 0.0120 | 0.17% | 0.0517 |

# G  (SEMI) SELF-SUPERVISED OPTIMAL POWER FLOW

Figure 5 shows that, even without access to exact solver outputs, incorporating a sparse set of solver sensitivities into the loss markedly improves feasibility: the median reduction in the *maximum* constraint violation on the `ieee300` test set exceeds 9 %. Table 8 confirms that this gain comes at the cost of only a modest increase in optimality gap (1.65 % versus 0.56 %) and mean-squared error (0.016 versus 0.010). Both gaps remain below the 2 % threshold, whereas infeasibility is reduced by an order of magnitude.

The experiment also reiterates the importance of the sparse Jacobian mask: the 95 % sparsity level identified in Appendix F was essential for stable training. Overall, these findings support the main-paper conclusion that even partially available first-order information can steer a proxy toward solutions that are far more reliable for safety-critical AC-OPF workloads.

Table 8: Average performance of OPF Proxy Models (Sobolev vs MSE).

| Case | Metric | Sobolev | Bench |
|---|---|---|---|
| `ieee300` | MSE | 0.016 | **0.010** |
| `ieee300` | GAP | 1.65% | **0.56%** |
| `ieee300` | INF | **0.0011** | 0.012 |

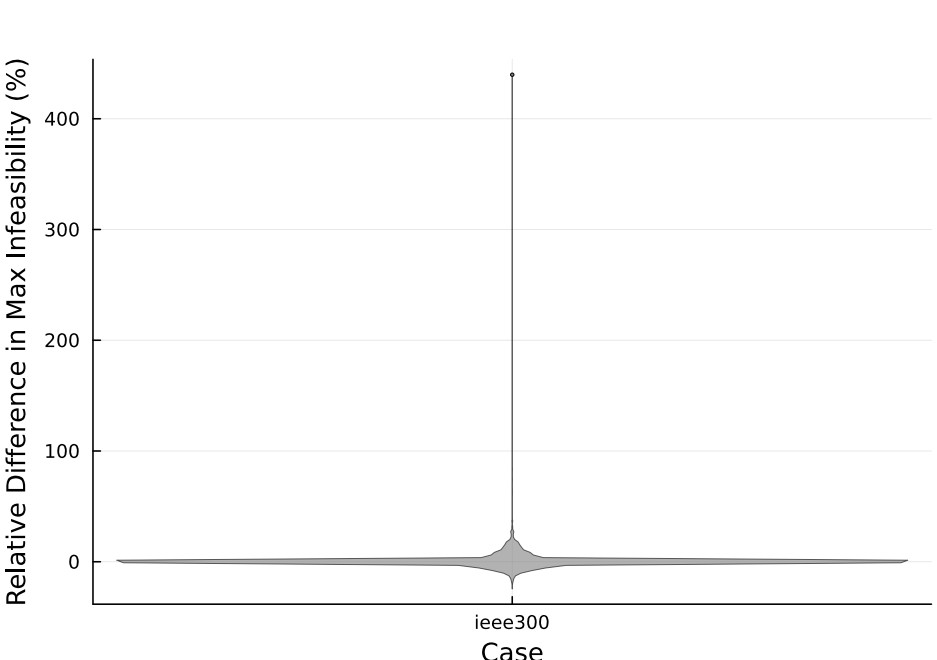

Figure 5: Relative max Infeasibility Difference (%).

