# OpenReview forum: "Sobolev Training of End-to-End Optimization Proxies"
_ICLR.cc/2026/Conference — Submitted to ICLR 2026_

### Official Review · Reviewer_X1dD · 2025-10-25

**Soundness:** 2
**Presentation:** 2
**Contribution:** 2
**Rating:** 6
**Confidence:** 1

**Summary:**

The paper proposes Sobolev Training of End-to-End Optimization Proxies: neural networks that amortize parametric optimization by approximating the solution operator. The key idea is to supervise both values and solver sensitivities (first-order derivatives) via a sparsity-masked Sobolev loss.

**Strengths:**

Learning reliable optimization proxies with good gradients is highly relevant for nested decision pipelines and time-critical applications. The sparsity-masked Jacobian supervision is a practical and nontrivial twist.

**Weaknesses:**

The choice of mask density (e.g., 5–25%, often ~95% sparsity) is tuned empirically. There’s limited guidance or theory for which Jacobian entries to supervise beyond random sparsification.

The theoretical guarantees assume exact, dense Jacobians, but experiments use noisy, sparse, and occasionally invalid sensitivities.

Theoretically, matching derivatives should reduce sample complexity, but the paper provides no data-efficiency curves (performance vs. number of training samples).

The baseline coverage seems limited.

**Questions:**

See Weakness

---

> ### Author Response · Authors · 2025-11-19
>
> We thank Reviewer 4 for the careful reading and constructive feedback. We hope our responses address the points raised and clarify the paper’s contributions.
>
> > The theoretical guarantees assume exact, dense Jacobians, but experiments use noisy, sparse, and occasionally invalid sensitivities.
>
> We acknowledge that the bounds are stated for exact, dense Jacobians. In training, however, we use randomly masked vector–Jacobian products with a simple reweighting, which forms an unbiased Monte Carlo estimator of the dense Sobolev objective (cf. [1]). Thus, sparsity affects the variance of the estimator, not its target, and does not impede convergence in expectation for first-order methods.
>
> We obtain sensitivities via auto/implicit differentiation and supervise only the projected components selected by the mask. This projection stabilizes learning; any noise manifests as higher variance rather than bias in the gradient estimate. Empirically, we observe stable convergence and the feasibility improvements reported in the paper, consistent with the theoretical motivation.
>
> > Theoretically, matching derivatives should reduce sample complexity, but the paper provides no data-efficiency curves (performance vs. number of training samples).
>
> We agree that, in principle, matching derivatives should reduce sample complexity and that data-efficiency curves are informative. However, such curves presuppose that the benchmark can reach performance comparable to Sobolev, especially on feasibility. In practice, learning high-quality proxies is difficult: despite universal approximation, value-only training typically plateaus and its validation performance diverges, suggesting a challenging optimization landscape. As a result, we were unable to bring the benchmark to a similar feasibility level, making head-to-head sample-size sweeps less meaningful. If a value-only approach could attain comparable feasibility, a data-efficiency curve would be useful; to date, we have not observed such a competitor.
>
> > The baseline coverage seems limited.
>
> We acknowledge the concern. Our intent was to evaluate training losses rather than architectures or post-processing. Accordingly, we compared against the canonical value-only baselines used in the optimization-proxy literature: supervised training with MSE and self-supervised training with a penalized Lagrangian loss. Many alternative “baselines” in prior work involve projection layers or architectural choices that are orthogonal to the loss and can be composed with either approach; our study isolates the effect of Sobolev supervision under a fixed architecture. Where a closed-form projection is available (MPS), it is applied identically to all methods; where it is not (AC-OPF), none is used. Thus, the coverage targets the standard loss-level baselines while keeping comparisons fair and focused.
>
> > References
>
> [1] Wojciech M Czarnecki, Simon Osindero, Max Jaderberg, Grzegorz Swirszcz, and Razvan Pascanu. Sobolev training for neural networks. Advances in neural information processing systems, 30, 2017.

---

### Official Review · Reviewer_z9BG · 2025-10-31

**Soundness:** 2
**Presentation:** 3
**Contribution:** 2
**Rating:** 4
**Confidence:** 4

**Summary:**

This paper introduces Sobolev Training of End-to-End Optimization Proxies, a framework for learning differentiable surrogate models that approximate the solution map of parametric optimization problems. Instead of training solely on optimal decisions, the proposed method incorporates the Sobolev loss, which matches both function values and gradients, allowing the mapping function to achieve better convergence to the original landscape. In theory, the paper formalizes conditions under which the optimal solution mapping is differentiable and proves that it yields a quadratic improvement in approximation error. The experiments include AC Optimal Power Flow and Mean–Variance Portfolio Optimization, which demonstrate improved feasibility and generalization, although the numerical gains are sometimes modest.

**Strengths:**

The paper is conceptually original in combining Sobolev training with the learning of end-to-end optimization proxies. This bridges a clear gap between differentiable optimization and learning-based surrogate modeling. The idea of using solver sensitivities or implicit KKT derivatives as gradient supervision is elegant and well-motivated.

On the technical side, the theoretical analysis is rigorous and clearly articulated. The authors carefully state the regularity assumptions (LICQ, SOSC) and derive a clean result showing a quadratic improvement in approximation error for value-plus-gradient training. The connection between the Sobolev loss and the smoothness of the solution map is intuitive and theoretically grounded.

The sparse Jacobian masking and projection layer are practical engineering contributions that make Sobolev training computationally feasible.

Overall, the paper is clear, technically sound, and relevant to the growing field of learning to optimize.

**Weaknesses:**

While the paper is theoretically sound and conceptually elegant, its main limitation lies in the modest empirical improvement. The quantitative gains over value-based baselines are small, and the claimed advantage in improving feasibility is not clearly demonstrated. The authors emphasize feasibility as a key benefit, yet the current metrics only partially capture this aspect. If feasibility improvement is indeed the main contribution, the paper would benefit from more comprehensive metrics and visualizations, such as the proportion of feasible instances or the number of violated constraints. At present, the reported metrics have limited explanatory power.

Furthermore, if feasibility is considered the main contribution of the paper, the evaluation should include comparisons with other feasibility-oriented approaches, such as projection-based methods. In the Mean–Variance Portfolio Optimization experiment, the authors additionally apply a projection layer to enforce feasibility but do not report results without this layer or provide any ablation analysis. As a result, it is difficult to disentangle whether the observed improvements come from the Sobolev loss itself or from the feasibility projection. This omission weakens the central claim that Sobolev training inherently improves feasibility.

The experimental design is confusing and inconsistent across tasks. AC-OPF and portfolio optimization each have both supervised and self-supervised variants, but these are split between the main text and appendices without a unified comparison. A summary table comparing all four settings (task × training × projection × mask) would greatly clarify the experimental logic. Moreover, portfolio experiments include a feasibility projection layer while AC-OPF does not, making cross-task comparisons uneven.

**Questions:**

1. For self-supervised learning, how are the sensitivities or partial Jacobians obtained without ground-truth optimal labels?
2. nested-AD and higher-order derivatives are mentioned as very costly. Please report wall-clock and GPU memory vs. value-only training, and any stabilization tricks (such as mask).
3. How is the binary mask constructed?
4. Provide an ablation over mask density (e.g., 5/10/25/50%) and show its effect on efficiency and performance.
5. In the portfolio optimization task, both baselines use a projection layer, while the AC-OPF experiments do not. Could you report results without the projection layer and clarify why this asymmetry exists across tasks?

---

> ### Author Response · Authors · 2025-11-19
>
> We thank Reviewer 3 for the careful review. In response, we clarified key assumptions and added missing experimental details - we hope it meets with the reviewer’s expectations.
>
> > Weakness – Contribution and Feasibility Importance
>
> Consistent with the optimization-proxy literature, we assume a decision maker who solves a constrained problem but must rely on a surrogate at inference time due to strict latency. In this setting, feasibility is the primary concern, with small, bounded deviations tolerated when unavoidable (as in classical numerical solvers). This is especially true for constraints beyond trivial bounds or those enforced by construction (e.g., physical equalities such as robotic dynamics or steady-state power flow), where limited tolerances are often operationally acceptable.
>
> Our contribution is a framework for training fast proxies—surrogates with single-pass inference—that (i) comes with tighter approximation guarantees, and (ii) delivers either better objective performance (e.g., the MPS case under medium risk budgets) or substantially improved feasibility relative to value-only proxies, typically with only a small objective trade-off.
>
> Training optimization proxies is intrinsically challenging. Our Sobolev approach mirrors classical constrained optimization by emphasizing feasibility first, then optimality. The observed gains in feasibility—both on average and in worst-case tails—are large and consistent with this principle.
>
> > Furthermore, if feasibility is considered the main contribution of the paper, the evaluation should include comparisons with other feasibility-oriented approaches, such as projection-based methods.
>
> We agree that feasibility is an important axis of evaluation. Projection layers act at inference by repairing outputs, whereas our Sobolev loss acts at training by aligning the proxy with solver sensitivities and improving the Lagrangian approximation (which includes feasibility). The two are complementary, not exclusive—indeed, in our MPS setting, a projection step was composed with Sobolev training.
>
> Although beyond the scope of this work, since projections are often most effective when network predictions are already near-feasible [6–8], a possible extension would be to further analyse how Sobolev training could improve the effectiveness of projection/repair layers applied at inference.
>
> Nevertheless, for general constraint sets there is no inexpensive projection: computing it often amounts to solving a constrained optimization problem. Closed form or cheap projections exist only for special structures (e.g., boxes, \ell_2 balls, ...). Relying on projection at inference can therefore erase the latency advantage of learned proxies in many realistic problems. That is why we present no projection step for AC-OPF.
>
> Our goal is to reduce violations without per-instance solves. Sobolev supervision improves feasibility pre-projection, preserves single-pass inference, and can still be paired with a projection (when cheap and available) as a safety net. We will clarify this relationship in the paper and, where structure admits an efficient projection, note that it can be added without changing our core contribution.
>
> Reference
>
> [6] Chen, Y., Huang, D., Zhang, D., Zeng, J., Wang, N., Zhang, H., & Yan, J. (2021). Theory-guided hard constraint projection (HCP): A knowledge-based data-driven scientific machine learning method. Journal of Computational Physics, 445, 110624.
>
> [7] Chen, H., Flores, G. E. C., & Li, C. (2024). Physics-informed neural networks with hard linear equality constraints. Computers & Chemical Engineering, 189, 108764.
>
> [8] Min, Y., & Azizan, N. (2024). Hard-constrained neural networks with universal approximation guarantees. arXiv preprint arXiv:2410.10807.
>
> Suggested Change Section 3 [End Paragraph 4 - Feasibility]:
>
> ```
>
> Accordingly, feasibility projections and Sobolev training are complementary rather than competing; and for general constraint sets without computationally efficient projections, Sobolev supervision offers a significantly more efficient way to improve feasibility.
>
> ```

---

> ### Author Response · Authors · 2025-11-19
>
> Response Continued:
> > In the Mean–Variance Portfolio Optimization experiment, the authors additionally apply a projection layer to enforce feasibility but do not report results without this layer or provide any ablation analysis. As a result, it is difficult to disentangle whether the observed improvements come from the Sobolev loss itself or from the feasibility projection. (& Question 5)
>
> We appreciate the concern about disentangling the effects of the projection from those of the Sobolev loss. In the MPS experiment the projection is a closed-form, inexpensive map (a simple rescaling to satisfy the variance cap and budget). We apply the identical projection to both methods (Bench and Sobolev), so any difference in outcomes arises from the training signal only, not the projection itself: the projection is a deterministic post-processing step independent of how the proxy was trained.
>
> > The experimental design is confusing and inconsistent across tasks. AC-OPF and portfolio optimization each have both supervised and self-supervised variants, but these are split between the main text and appendices without a unified comparison. A summary table comparing all four settings (task × training × projection × mask) would greatly clarify the experimental logic. Moreover, portfolio experiments include a feasibility projection layer while AC-OPF does not, making cross-task comparisons uneven. (& Question 5)
>
> The goal of the experiments was to show that Sobolev is general and can be used in different training regimes. Our experimental design intentionally includes one supervised setting and one self-/semi-supervised setting to demonstrate that the framework improves the decision-relevant objective—the Lagrangian (objective plus penalties for constraint violation)—regardless of supervision regime. We do not aim to conduct cross-task comparisons or to assert that Sobolev training yields distinct effects in supervised versus self-supervised settings. The impact of Sobolev training necessarily depends on the application, architecture, and data. The key takeaway is compatibility with a decision maker’s preferences: behaviour aligned with classical constrained solvers while retaining extremely fast inference, which is precisely the motivation for using optimization proxies.
>
> > Question 1
>
> Although solution sensitivities can be obtained via implicit differentiation of the proxy’s predictions (what would be a fully self-supervised approach), in our (semi) self-supervised experiments the instances were small enough that, after roughly half of the training epochs—the time needed to solve all instances in the dataset—we transitioned from value only learning to using the labelled optimal solutions to compute sensitivities.
>
> In the MPS study, the self-supervised variant substantially outperforms supervised baselines, which is the reason for its choice, even though the sensitivities were computed from true labels. Accordingly, using labels solely to form sensitivity targets does not contradict our premise; it is a pragmatic, performance-oriented choice that preserves the benefits of self-supervision. Nevertheless, for industry-scale applications, ground-truth labels can be prohibitively expensive to obtain, which would motivate a fully self-supervised training.
>
> > Suggested Change Section 6 [End of Paragraph 3 – Training]
> ```
> For the Sobolev approach, after roughly half of the maximum number of training epochs (20K)—the time needed to solve all instances in the dataset—training transitioned from value only learning to using the labelled optimal solutions to compute sensitivities.
> ```
> > Suggested Change Section 7 [New Last Paragraph]
> ```
> In the MPS study, the self-supervised variant substantially outperforms supervised baselines, which is the reason for its choice, even though the sensitivities were computed from true labels. Accordingly, using labels solely to form sensitivity targets does not contradict the premise; it is a pragmatic, performance-oriented choice that preserves the benefits of self-supervision. Nevertheless, for industry-scale applications, ground-truth labels can be prohibitively expensive to obtain, which would motivate a \textbf{fully} self-supervised training. In turn, this would require that the implicit function differentiation be computed in \textbf{GPU} and in \textbf{batch} from proxy’s predictions.  Although recent progress in GPU batch-compatible linear-system solvers (e.g., cuDSS) is promising, further improvements in numerical stability and precision are needed to ensure reliable performance.
> ```

---

> > ### Author Response · Authors · 2025-11-19
> >
> > Response Continued:
> > > Question 2
> >
> > All methods were trained for 20K epochs, and the measured average epoch times (seconds) show a modest and bounded overhead during training only:
> >
> > SUPERVISED OPF (300 Bus)
> >
> >  - MSE loss benchmark: 0.2892s
> >
> >  - Sobolev: 0.6312s
> >
> > SUPERVISED OPF (1k Bus)
> >
> >  - MSE loss benchmark: 0.4369s
> >
> >  - Sobolev: 0.9593s
> >
> > SUPERVISED OPF (6k Bus)
> >
> >  - MSE loss benchmark: 0.9625s
> >
> >  - Sobolev: 1.5551s
> >
> > (SEMI) SELF-SUPERVISED OPF (300 Bus)
> >
> >  - Benchmark: 0.6396s
> >
> >  - Sobolev: 0.7714s
> >
> > SELF & SUPERVISED MPS
> >
> >  - Benchmark: 0.0941s
> >
> >  - Sobolev: 0.2432s
> >
> > Although Sobolev models entail longer training times, we did not optimize wall-clock performance given the offline setting. For the largest OPF case, we chose a 25% sparse mask to prioritize accuracy; the only practical constraint was that total training time remain under one day. In the revised manuscript, the training-time table has been added to the Appendix and is cited in the methodology for transparency.
> >
> > > Question 3
> >
> > The proposed approach requires an unbiased binary stochastic mask ($M$) with target density to select entries for vector–Jacobian products. Julia’s `SparseArrays.sprand` was a perfectly suitable way to sample the support of $M$.
> >
> > > Question 4
> >
> > We report an ablation over mask density in the appendix (due to page limits). As hinted in the limitations’ discussion of full-Jacobian costs, the efficiency impact scales with density. The appendix table with the 25% mask illustrates this trend; additional ablations would reiterate the same relationship.
> >
> > On performance, densities in the 5–25% range consistently produced the best proxies in our experiments. Tuning within this interval can yield marginal gains, but we recommend such hyperparameter search only when the extra training compute is available.
> >
> > > Question 5
> >
> > In the portfolio (MPS) study, the projection is a closed-form, inexpensive rescaling to meet the variance cap and budget, and it is applied identically to both Bench and Sobolev; hence any performance gap reflects the training signal, not the projection. Running without this projection simply yields larger raw violations for both models and does not change the interpretation (Sobolev remains more feasible). The asymmetry with AC-OPF arises because there is no comparably cheap projection for physical network constraints—adding one would effectively require re-solving an OPF subproblem, defeating the latency goal of proxies—so AC-OPF is evaluated without projection.

---

### Official Review · Reviewer_a5bU · 2025-11-01

**Soundness:** 2
**Presentation:** 3
**Contribution:** 2
**Rating:** 2
**Confidence:** 3

**Summary:**

The paper introduces a Sobolev training framework for learning neural optimization proxies that jointly match solution values and the solver’s geometric sensitivities. To improve scalability, a sparse masking strategy is proposed to reduce the memory overhead of full Jacobian supervision and mitigate gradient conflicts in large-scale constrained learning. The authors derive uniform approximation bounds for value-only, Jacobian-only, and joint Sobolev training, and demonstrate that the proposed method outperforms value-only matching in both supervised and self-supervised settings.

**Strengths:**

- Incorporating solver sensitivities into the loss provides a conceptually intuitive way to teach the network about local geometric structure. The proposed sparse masking strategy is simple, effective, and well-motivated: it reduces memory usage, mitigates gradient conflicts, and ablation studies demonstrate that high sparsity levels (keeping only 5–25% of entries) yield the best performance.

- Experimental results show consistent reductions in mean squared error and, notably, in worst-case infeasibility and outliers, which is particularly valuable for safety-critical applications.

- The paper is clearly written, well-organized, and self-contained.

**Weaknesses:**

- While the paper provides bounds for value-only, Jacobian-only, and joint Sobolev training individually, there is no theoretical guarantee that joint Sobolev training consistently achieves a smaller error gap than value-only training.

- Although Sobolev training substantially improves MSE and infeasibility metrics compared to the benchmark, the opposite trend is observed for the optimality gap. This suggests that Sobolev may produce solutions that are close to optimal in Euclidean space but not necessarily in the function space (e.g., in terms of surface or contour).

**Questions:**

- What is the core novelty of this work compared to traditional Sobolev training? Is the primary contribution the masking mechanism M? If the main innovation is limited to the masking strategy, it may be considered incremental, and its novelty could be questioned. It would be helpful to clarify how this approach meaningfully extends or improves upon prior Sobolev-based methods.

- Incorporating the gradient of the proxy into the loss function may necessitate computing the Hessian during model updates. This can become computationally expensive, particularly when the proxy has a complex architecture with a large number of parameters. The issue may be further exacerbated when the underlying solver itself is complex, as representing its behavior accurately would require an even larger and more expressive proxy.

- Empirically, the benchmark (Bench) achieves a significantly better optimality gap while potentially requiring lower training cost compared to Sobolev training. Given this, it is unclear under what circumstances one would prefer Sobolev over Bench in practice.

- Minor points: Figure 2 is not referenced in the main text. Additionally, in lines 308 and 322, the term “Model” should perhaps be replaced with “Figure.”

---

> ### Author Response · Authors · 2025-11-19
>
> We thank Reviewer 2 for taking the time to review our submission and for providing detailed comments. We appreciate the opportunity to clarify our methodology and strengthen the presentation of our work in light of the reviewer’s observations.
>
> > Weaknesses
>
> No reasonable framework, training method, loss function, or post-processing step can guarantee uniformly superior performance for surrogate models across all problems and metrics against other reasonable alternatives. At best, we can provide probabilistic or asymptotic guarantees that motivate particular choices, which then must be validated empirically in each setting.
>
> Our theoretical results extend prior work by proving a tighter approximation bound, and our experiments illustrate both the usefulness and the limits of the proposed loss. In practice, Sobolev training for optimization proxies markedly improves feasibility (and reduces outliers) at a small cost in optimality gap and with a moderate increase in training-time compute. This pattern is consistent with a Lagrangian view of constrained optimization: violations of constraints carry substantially larger effective penalties than small changes in objective value, so aligning the neural network’s local geometry (via derivative supervision) encourages satisfaction of constraints, even if it moves solutions slightly along objective level sets.
>
> Under the premise—shared by the literature on optimization proxies [4,5]—that the decision maker ultimately optimizes a constrained problem, the operative objective is the Lagrangian (or a closely related penalized form). In that regime, the observed trade-off is warranted: prioritizing feasibility and safety is often preferable to marginal improvements in objective value. Sobolev training therefore brings the surrogate closer, in function space, to the mapping that optimizes the Lagrangian, which is typically the relevant target for real deployments.
>
> In light of the reviewers’ comments, the Contributions section now opens each item with a concise summary:
>
> > Suggested Change Section 1 – Contributions
>
> ```
>
>  - A novel sparse-masked Sobolev loss that supervises a selected subset of sensitivities via unbiased vector–Jacobian products—avoiding Hessians, mitigating gradient interference, and scaling stably on GPU to large constrained problems while preserving single-pass inference.
>
>  - New uniform value-and-gradient approximation guarantees that extend existing error bounds and supports usage of Sobolev training for constrained $\argmin$ maps.
>
>  - An end-to-end pipeline that generates diverse instances and exact KKT sensitivities, applies a task-aware sparsity mask, and trains multi-thousand-variable proxies efficiently on a single GPU, with reproducibility details.
>
>  - At modest offline training overhead and unchanged inference cost, empirical results across AC-OPF studies show large reductions in worst-case infeasibility and mean–variance portfolio case supports an absolute dominating mixture-of-experts strategy.
>
> ```
>
> > Question 1
>
> Under the premise that a decision maker solves a constrained optimization problem and seeks a surrogate for that mapping, our work is—to our knowledge—the first to adapt Sobolev supervision specifically to optimization proxies. The contribution is not limited to the masking mechanism M. The other key novelty is aligning supervision with the solver’s sensitivities/KKT geometry (i.e., the Jacobian of the argmin map with respect to context), rather than generic input–output gradients used in traditional Sobolev training. This alignment targets the quantities that determine feasibility and constraint activity, which are central in proxying constrained solvers.
>
> The sparse mask is an enabling component that makes this supervision tractable in the high-dimensional, constraint-heavy setting typical of optimization proxies and helps mitigate gradient interference; it is not the end in itself. Coupled with our analysis and empirical studies, the approach yields a principled and scalable Sobolev framework tailored to learned optimization proxies—where the dimensionality and structure of the sensitivity information warrant the proposed sparsity.
>
> > Question 2
>
> Our aim is to approximate the optimization problem’s argmin map, not the internals of any particular numerical solver. By focusing on the problem level, we differentiate via optimality conditions (e.g., KKT/implicit function theorem) rather than performing fragile, solver-specific backpropagation. Modern differentiable optimization layers [3] provide efficient access to first-order solution sensitivities, which makes this feasible in practice from the dataset creation side. In our Sobolev loss, we never form full Jacobians by using the vector-product of the Jacobian with the random mask vector. This avoids the memory and compute overhead of Hessians while preserving the geometric information that drives feasibility and constraint satisfaction.

---

> > ### Author Response · Authors · 2025-11-19
> >
> > Response Continued:
> > > Question 3
> >
> > Under the premise—argued above—that the decision maker effectively optimizes a Lagrangian (prioritizing feasibility, with substantial penalties on constraint violation, over optimality), the observed trade-off is compelling: Sobolev training delivers large reductions in infeasibility while keeping the optimality gap small. This is precisely the regime where constraint satisfaction and safety dominate marginal objective improvements. The additional training cost is modest and incurred offline; inference remains a single forward pass.
> >
> > We do not claim universal superiority—no training framework can dominate a reasonable benchmark across all problems and metrics. Rather, we offer theoretical motivation and empirical evidence indicating when Sobolev is advantageous, and we encourage practitioners to evaluate it on their own instances.
> >
> > > Question 4
> >
> > Good point, AC-OPF model is now correctly labelled in the new paper version.
> >
> > > References
> >
> > [3] Rosemberg, A. W., Garcia, J. D., Pacaud, F., Parker, R. B., Legat, B., Sundar, K., ... & Van Hentenryck, P. (2025). A General and Streamlined Differentiable Optimization Framework. arXiv preprint arXiv:2510.25986.
> >
> > [4] Pascal Van Hentenryck. Optimization learning. arXiv preprint arXiv:2501.03443, 2025.
> >
> > [5] Brandon Amos et al. Tutorial on amortized optimization. Foundations and Trends® in Machine Learning, 16(5):592–732, 2023.

---

### Official Review · Reviewer_EXMc · 2025-11-02

**Soundness:** 3
**Presentation:** 3
**Contribution:** 3
**Rating:** 6
**Confidence:** 3

**Summary:**

The paper proposes a Sobolev training framework for solution prediction for optimization problems. Typically, when one trains a neural network to predict the optimal solution, it uses the solution value to guide the training. The framework studied in this paper also considers the accuracy of predicting the Jacobians of the solution, therefore the blackbox solution prediction function match both the solution and the local geometry of the solution space.

The framework contributions some novel techniques such as Sparse Sobolev loss, which makes training more efficient and tractable. It also provides theoretical analysis on the error bounds.

Empirically, the framework is evaluated in two real-world domains to demonstrate its effectiveness.

**Strengths:**

1. This is one of the first works in ML for optimization that uses Solobev training techniques. I consider this as an innovation for the research field. The sparse masking technique is novel and practically impactful.
2. Theoretical analysis shows that it helps with reducing sample complexity and improve generalization.
3. Empirical evaluation on two real-world domain demonstrates the effectiveness of the framework.

**Weaknesses:**

1. Jacobian computation in the framework produces extra overhead during training. The authors discussed how the overhead could be reduced, but it is still unclear how it compares to the benchmark method.
2. The results for portfolio optimization is not promising from a practical point of view. Since most funds / asset management firms in real world operates under tight risk in their portfolio management. Overall, portfolio optimization is not the best use case for studying optimization proxies, since rebalance frequencies are typically quite low when the covariance matrix is assumed to be estimated over long horizons.
3. Some experimental setups are not described in details such that the results could be reproducible. For example, how do you generate the instances for the MPS application? What are the hyperparameters used in training?

**Questions:**

1. How much overhead does your framework produce compared to the baseline during training?
2. What are the problem sizes of the MPS instances?

---

> ### Author Response · Authors · 2025-11-19
>
> We thank Reviewer 1 for the thoughtful assessment and constructive feedback. We hope that our clarifications and proposed revisions adequately address the reviewer’s concerns and strengthen the overall manuscript.
>
> > Weakness 1 & Question 1:
>
> We agree that naively forming full Jacobians would be prohibitive – an overall problem with any Sobolev method over Neural Networks (not just our proposed use for Optimization Proxies). However, our approach only requires vector–Jacobian contractions along a sparse, randomized mask, which substantially reduces the cost while preserving informative sensitivities – an extension of the vector product solution also employed in [1].
>
> Measured average epoch times (seconds) show a modest and bounded overhead during training only:
>
> SUPERVISED OPF (300 Bus)
>
>  - MSE loss benchmark: 0.2892s
>
>  - Sobolev: 0.6312s
>
> SUPERVISED OPF (1k Bus)
>
>  - MSE loss benchmark: 0.4369s
>
>  - Sobolev: 0.9593s
>
> SUPERVISED OPF (6k Bus)
>
>  - MSE loss benchmark: 0.9625s
>
>  - Sobolev: 1.5551s
>
> (SEMI) SELF-SUPERVISED OPF (300 Bus)
>
>  - Benchmark: 0.6396s
>
>  - Sobolev: 0.7714s
>
> SELF & SUPERVISED MPS
>
>  - Benchmark: 0.0941s
>
>  - Sobolev: 0.2432s
>
> This cost is paid only at training time. Inference remains a single forward pass (i.e., no Jacobian computation), which is the central motivation for optimization proxies, as highlighted in the abstract and introduction of the manuscript.
>
> > Suggested Change Section 7 [End of Paragraph 1]:
>
> ```
>
> Across our settings, the increases epoch time from the sparse derivative matching was approximately $1.2\times$--$2.6\times$ compared to the benchmark as shown in \ref{tab:train-overhead}. Nevertheless, this cost is paid only at training time, while inference remains a single forward pass (i.e., no Jacobian computation).
>
> ```
>
> > Weakness 2:
>
> We appreciate the reviewer’s concern. Many firms impose tight risk constraints in mean–variance models as a pragmatic hedge against distributional ambiguity in returns: given only a partially observed stochastic process, multiple distributions can explain the same history. In principle, this ambiguity is better handled within a distributionally robust optimization (DRO) framework so that decisions enjoy provable regret decay [2]. In practice, however, context-conditional DRO reduces to large complex programs with unfavourable scaling, making it impractical for high-frequency rebalancing.
>
> As a practical alternative to a fixed, tight risk budget, we advocate context-adaptive risk constraints—while recognizing that repeatedly solving even mean–variance programs under strict latency can still be onerous (hence the industry’s reliance on signal-based closed-form decision rules). Our proposal is therefore to learn a fast surrogate that proximally solves the portfolio problem, preserving general objective terms and constraints while enabling real-time use.
>
> We emphasize that our portfolio study is a proof-of-concept of the fundamental method rather than a full production specification for specific applications. A deployment-ready model would typically incorporate transaction costs, turnover and additional risk overlays, multiple covariance estimates, and (distributionally) robust terms. High-frequency decisions sometimes consult longer-horizon estimates (as opposed to hard-to-estimate intraday ones), but indeed those signals can be weak within intraday regimes; practitioners commonly blend horizons with ad hoc rules. In this spirit, our σ parameter is an illustrative device showing how the risk constraint can be incrementally relaxed, allowing the learned policy to traverse the performance–risk Pareto frontier in a controlled, context-aware manner.
>
> > Suggested Change Section 6 [Beginning Paragraph 3 - Motivation]:
>
> ```
>
> Practitioners often impose tight risk budgets in mean--variance models as a pragmatic hedge against \emph{distributional ambiguity} in returns: multiple data-consistent distributions can explain the same history. In principle, this ambiguity is best handled with distributionally robust optimization (e.g., \cite{van2021data}), but context-conditional DRO often reduces to large complex programs with unfavourable scaling, making it impractical for high-frequency rebalancing. As a practical alternative to a fixed, tight risk budget, the above-described version of the mean-variance portfolio proposes context-adaptive risk constraints.
>
> ```

---

> > ### Author Response · Authors · 2025-11-19
> >
> > Response Continued:
> >
> > > Weakness 3 & Question 2:
> >
> > To evaluate the approach on arbitrary asset universes rather than a fixed pool, we drew mean vectors uniformly from the ℓ₂ unit ball and generated positive semidefinite covariances via two complementary procedures:
> >
> >  - Random dense construction. Sample $A \in [0,1]^{n\times n}$ i.i.d. and set $\Sigma = A^\top A$.
> >
> >  - Factor-model construction. Simulate returns with Gaussian idiosyncratic terms and a small number of Gaussian common factors. For each 10k-sample history, factor loadings were randomly scaled in $[0, 0.3]$, and each factor influenced a random fraction of assets between $10%$ and $100%$. We then computed the empirical covariance $\Sigma$.
> >
> > In both cases, we verified symmetry/Hermiticity and positive definiteness to avoid numerical issues. The factor-model route was used to diversify the range of covariance structures without aiming to calibrate a fully realistic market model; it was not required for spanning PSD matrices but helped enrich the dataset. The portfolio dimension was n = 500.
> >
> > The above description was added to a new appendix and linked in the training strategy paragraph on Section 6.
> >
> > > References
> >
> > [1] Wojciech M Czarnecki, Simon Osindero, Max Jaderberg, Grzegorz Swirszcz, and Razvan Pascanu. Sobolev training for neural networks. Advances in neural information processing systems, 30, 2017.
> >
> > [2] Van Parys, B. P., Esfahani, P. M., & Kuhn, D. (2021). From data to decisions: Distributionally robust optimization is optimal. Management Science, 67(6), 3387-3402.

---

### Author Response · Authors · 2025-11-19

We thank all reviewers for their thoughtful assessments and constructive feedback. We clarified our problem framing (optimization proxies targeting the constrained argmin/Lagrangian), detailed how masked vector–Jacobian products provide unbiased Sobolev supervision without forming Hessians, and explained the complementary role of feasibility projections. We added missing experimental details (e.g., MPS data generation), corrected minor issues (figure references/terminology), and included a training-time table in the appendix, along with a concise summary of mask-density behaviour and its efficiency–accuracy trade-offs. Across OPF and MPS, the results consistently show large feasibility gains and reduced outliers at modest training-time overhead with unchanged inference cost. We believe these clarifications strengthen the manuscript and make the scope and contributions precise.

---

### Meta-Review · Area_Chair_cwAt · 2026-01-06

**Summary:**

The paper focuses on ML for optimization, where a neural surrogate is trained to approximate the solution of a parameterized constrained optimization problem in a single forward pass, yielding a fast proxy. It considers two commonly studied settings: (1) fully supervised training via regression to solutions produced by a classical solver, and (2) self-supervised training that evaluates the predicted solution through its optimality and feasibility gaps and minimizes these gaps directly. The core contribution is to train these neural proxies using a Sobolev-style loss (Czarnecki et al., NeurIPS 2017), i.e., augmenting value/solution matching with derivative information, implemented efficiently via random masking. The method is evaluated on AC Optimal Power Flow (OPF) and a mean–variance portfolio task.

Overall, I find the contribution fairly incremental. On the ML for optimization front, the problem setup and training regimes are standard, and the Sobolev-style loss (including the masking idea) follows prior work (Czarnecki et al., NeurIPS 2017). As a result, the main novelty is primarily empirical: demonstrating the effect of Sobolev-style training when applied to ML for optimization proxies on these benchmark tasks. Furthermore, given the limited breadth of optimization problems considered in the paper, it is not clear that the results support a broad conclusion that Sobolev-style losses significantly help across a wide variety of ML for optimization applications.

**Reviewer Concerns:**

The reviewers raised several concerns, some of which the authors addressed satisfactorily during the rebuttal. Key issues included:

* **Computational overhead and practicality of Jacobian/Sobolev supervision.**
* **Experimental clarity and consistency.** Reviewers found the setup confusing across tasks, with supervised vs. self-supervised variants split between the main text and the appendix.
* **Asymmetry due to projection layers.** The portfolio experiments use a feasibility projection layer, whereas AC-OPF does not; reviewers noted this complicates cross-task comparisons and asked for results without projection and/or a clearer justification.
* **Masking details and ablations.** Reviewers asked how the binary mask is constructed and requested ablations over mask density (e.g., 5/10/25/50%) to characterize the efficiency–accuracy tradeoff.
* **Reproducibility and missing experimental details.** Some experimental setups lack sufficient detail for reproduction (e.g., MPS instance generation, training hyperparameters, and problem sizes).
* **Portfolio task relevance and strength of results.** Some reviewers found the portfolio results less compelling in practice, noting that many real-world funds operate under tight risk constraints and that infrequent rebalancing may reduce the value of fast proxies.
* **Evaluation breadth and generality.** Concerns were raised that baseline coverage is limited and that, if feasibility is the main advantage, comparisons against other feasibility-oriented approaches (e.g., projection-based methods) would strengthen the evaluation.
* **Theory–empirics gap on data efficiency.** Although derivative matching is motivated as improving sample complexity, reviewers noted the absence of data-efficiency curves (performance vs. number of training samples).
* **Mismatch between theory and implementation.** The theoretical results assume access to the full Jacobian, while the proposed method uses a randomly masked approximation.

**Reviewer Scores:**

The reviewers’ scores were mixed: two rated the paper as marginally above the acceptance threshold, one marginally below, and one recommended rejection.

I recommend rejecting the paper for three reasons:

1. The contributions are limited.
2. The experimental breadth is insufficient to support a conclusive takeaway.
3. Several reviewers' concerns remain unresolved and would require substantial revisions.

---

### Decision · Program_Chairs · 2026-01-26

Reject